# Current knowledge of physicians' dual practice in Iran: A scoping review and defining the research agenda for achieving universal health coverage

**Javad Moghri**[1], **Jalal Arabloo**[2]*, **Mohammad Barzegar Rahatlou**[3], **Maryam Saadati**[3], **Negar Yousefzadeh**[3]

1 Social Determinants of Health Research Center, Mashhad University of Medical Sciences, Mashhad, Iran,
2 Health Management and Economics Research Center, Iran University of Medical Sciences, Tehran, Iran,
3 School of Health Management and Information Sciences, Iran University of Medical Sciences, Tehran, Iran

* arabloo.j@iums.ac.ir

**Data Availability Statement:** All relevant data are within the article and its Supporting information files.

## Abstract

### Background

Physicians' dual practice (simultaneous practice in both public and private sectors) may be challenging for achieving universal health coverage. The purpose of this review is to identify the types of available evidence in physicians' dual practice in Iran and define the research agenda for achieving universal health coverage (UHC).

### Methods

We conducted a scoping review of the literature using Arksey and O'Malley's approach. We searched Embase, PubMed, the Cochrane Library, Scopus, Web of Science core collection, as well as internal databases including the National Magazine Database (Magiran) and the Scientific Information Database (SID) until August 3, 2020. Studies published in Persian or English and investigating physicians' dual practice in the health system of Iran were included. Each step of the study was performed by two of the present researchers. The Preferred Reporting Items for Systematic Reviews and Meta-Analyses Extension for Scoping Reviews (PRISMA-ScR) recommendations were used to conduct this study and report the findings.

### Results

Fourteen studies were included in the current review. The findings were categorized and synthesized into five themes including the forms of dual practice, the extent of dual practice, the motivators and factors affecting dual practice, the policy options, and the consequences of dual practice. There were limited evidence on the nature, types, and prevalence of this phenomenon for different provinces and medical specialties and on health policy options in Iran. There seems to be a methodological gap (a gap in the type of study and its method) in

**Funding:** This research has been supported and funded by the National Institute of Health Research of I.R. Iran, Tehran University of Medical Sciences (Contract No: 9928). The funder played no role in the design of the study, in the collection, analysis, and interpretation of the data, and in writing the manuscript.

**Competing interests:** The authors have declared that no competing interests exist.

the subject area. Most studies have only used quantitative or qualitative study methods and based on the self-report of research samples in most of the included studies.

## Conclusions

More research is required at national level on the nature, types, and prevalence of this phenomenon, focusing on clarifying the root causes of this phenomenon and on the effects of dual practice on the indicators of accessibility to health services, especially for vulnerable populations, the quality of care provided, and equity, and on complex policy research on health policy options in Iran. The research questions proposed in the present study can help to bridge the knowledge gap in this area. Additional studies should address issues related to the quality of data collection in physicians' dual practice.

## Introduction

Physicians' dual practice is a common phenomenon in countries with a hybrid health system which is based on the participation of actors and stakeholders from both public and private sectors and potentially affects the access, equity, and quality of services. Dual Practice refers to holding jobs in two sectors at the same time. In point of fact, dual practice makes sense when a physician works simultaneously in private and public sectors, two public centers, or two private centers [1, 2].

The dual practice is double-edged in developing and developed countries; however, its negative effects are greater than the positive points. Favorable consequences include increasing the health workers' salary and retention rate, especially in deprived areas, and reducing the excess burden on the public sector budget [2].

In many countries, physicians are allowed to work simultaneously in the public and private sectors of the health care systems. Dual practice can be controversial in terms of universal health coverage; it is due to the fact that "Universal Health Coverage" aims to provide health care services based on "need" rather than the ability to pay. In mixed public-private healthcare systems, the time allotted to each sector according to physicians' preferences affects access to health services, including waiting times in public hospitals, as well as the costs and consequences associated with patients' health. Therefore, moving toward universal health coverage (UHC) requires sufficient health personnel to be available and also be motivated to do so. In relation to effective health workforce development, almost little attention has been paid to the trends and consequences of dual practice [3, 4].

In Iran, the dual practice of physicians in the public and private sectors is one of the main challenges of the health system. According to a study, the rate of the dual practice of physicians in the private sector is 42.3%. Although there is a complete ban on dual practice under the national law, such factors as the need to increase the share of health costs from public sources, as well as the lack of tariffs based on actual costs, have led to the dilemma of enforcing the law but not strictly following it [5]. Based on the foregoing, conducting a study with the aim of identifying the status of evidence in the field of dual practice in the Iranian health system is needed. This study pursued the following specific objectives:

- Recognizing the existing evidence in the field of dual practice in the Iranian health system.

- Identifying the characteristics of the included studies, such as author, type of publication, date, and language of publication.

- Certifying the concepts, characteristics, or factors related to dual practice in the Iranian health system and summarizing the findings of the studies.

- Determining the knowledge gaps in the field of the subject that affect universal health coverage.

## Materials and methods

The present scoping review was conducted to identify and summarize the available evidence and identify knowledge gaps in the mentioned area [6, 7]. In this study, the five-step method of Arkesy and O'Malley was followed. Developed in 2005, this framework includes five steps: formulating a research question, identifying relevant studies, selecting studies, extracting data, and reporting results [6].

The databases of Embase, PubMed, Cochrane Library, Scopus, and Web of Science core collection, as well as internal databases including the National Magazine Database (Magiran) and the Scientific Information Database (SID), were searched until August 3, 2020. The Google Scholar search engine as well as related websites was used to search for gray literature (S1 Table). Finally, a snowballing strategy was used to complete the search. In this way, the reference list scanning (Reference list scanning) and the scientific profile of researchers and specialists with related research (Author tracking) were further examined to identify related studies. In order to extract the maximum evidence related to the subject, searches of databases and gray literature were performed without any restrictions on the time of publication, language, or type of publication (original or review research article, report, dissertation, etc.).

A combination of keywords and MeSH terms was used to search the databases. A specific search strategy was developed for each of the above-mentioned databases. After removing duplicates, the title and abstract of the remaining studies were checked and irrelevant articles were excluded. The full texts of the remaining articles were reviewed based on the inclusion and exclusion criteria, and the reasons for their exclusion from the study were documented for each of the articles. All phases associated with the article selection process were undertaken by two independent researchers. Disagreements between reviewers were resolved by the decision of a third reviewer.

In the current study, dual practice meant the simultaneous employment of physicians in both private and public sectors. The studies were selected based on the following eligibility criteria:

### Inclusion criteria

1. All primary or secondary studies on dual practice in physicians at any level of the Iranian health system

2. Only the studies with a full text in English or Persian were included.

### Exclusion criteria

1. Studies on dual practice in other health professionals such as nurses

2. Review studies, editorials, letters to the editors, abstracts, and commentaries;

3. Studies without a full text or with a full text in a language other than English or Persian;

4. Duplicate studies whose findings have been published in several articles, and in this case, the most high-quality ones were included in the study.

Endnote X7 reference management software was used to manage references. After selecting the studies in accordance with the research questions, a specific form was designed for extracting the required information from them. Data extraction related to each of the entered studies was performed independently by two researchers and was controlled by a third person. Key information on the included studies, such as study characteristics, subjects, and main findings, was presented in the form of texts and tables. The extracted data were analyzed using qualitative synthesis. To perform the study and report the results, we followed the Preferred Reporting Items for Systematic reviews and Meta-Analyses Extension for Scoping Reviews (PRISMA-ScR) guidelines [8].

### Ethics statement

The present study was approved by Ethical Committee of the Tehran University of Medical Sciences (ethical approval reference number IR.TUMS.NIHR.REC.1399.004).

## Results

### Quantity of evidence

Fig 1. shows the flow diagram of the study. The results of the electronic search of the databases initially yielded 449 documents. After removing duplicates, 403 titles and abstracts were screened. After studying the full text of the studies, fourteen studies were included based on the inclusion and exclusion criteria.

### Characteristics of the included studies

Table 1 shows the characteristics of the included studies. The studies were published in Persian (n = 5) and English (n = 9). A mixed methods design was used, including quantitative (n = 8) and qualitative (n = 6) approaches. In qualitative studies, the sample size was in the range of 13 to 17 people, but in quantitative studies, there was a wide range of sample sizes (30 to 24414). The studies were conducted over a decade from 2010 to 2020. in 2010 (n = 1), 2013 (n = 1), 2017 (n = 1), 2020 (n = 1), 2015 (n = 1), 2019 (n = 4), and 2018 (n = 5). Most of the studies were first-authored by Bayat (n = 4).

### Main themes

Table 1 summarizes and presents details of the findings, divided into five sub-themes, namely the forms of dual practice, the extent of dual practice, the motivators and factors affecting dual practice, the policy options, and the consequences of dual practice.

**Forms.** Six studies examined the simultaneous employment of physicians in the public and private sectors [9–14]. Four studies were conducted to examine the employment of physicians and specialists in more than one place of service delivery [5, 10, 15, 16]. Two studies examined the various aspects of the obligation with respect to the full-time presence of faculty members [17] and full-time geographical specialists in the public sector [18], and two research studies also assessed the ban on the employment of public physicians in the private sector from a legal perspective [19, 20].

**Extent.** Five studies reported the rates of dual practice for general practitioners and medical specialists [5, 11, 12, 15, 21]. These rates were reported to be 18.65% for general practitioners working in both public or private sectors, 26.17% for general practitioners working in two

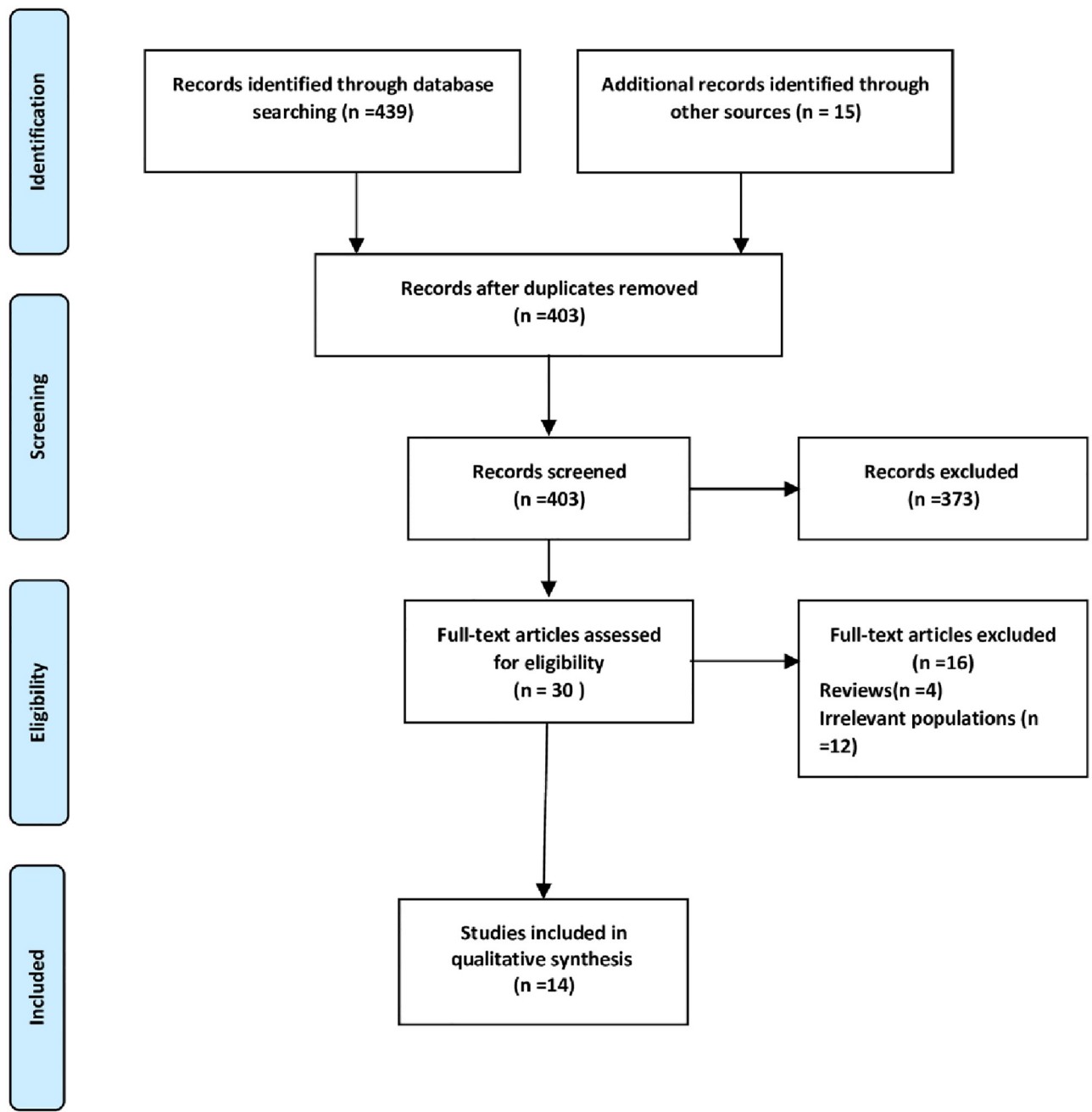

**Fig 1. The Preferred Reporting Items for Systematic Reviews and Meta-Analysis (PRISMA) flow diagram of the study selection process.**

or more job places (either public or private) [11], 57% for surgeons working in the public sector, 70% for non-full-time specialists in multiple job holdings, 58% for non-faculty member specialists in multiple job holdings, 69% for urinary tract and genital and neurosurgery [21], 63.1% for surgical specialists in other service locations [12], 47.7% for specialists in various medical specialties [15], and 42.3% for public medical specialists in the private sector [5].

In Iran, the rate of dual practice varies across provinces so that it was significantly higher in more populous and more deprived provinces and also in those which provide more private

Table 1. Characteristics of the included studies and their main findings.

| Author, year | Language | Aim | Study Method | Population | forms | Extent | Predictors, drivers and motivations | policy options | consequences for UHC goals |
|---|---|---|---|---|---|---|---|---|---|
| Yousefi & Taziki (2018) | Persian | To evaluate the attitude and function of full-time clinical faculty members in educational activities with optimal therapeutic services in Gorgan. | Qualitative study | 17 full-time faculty members | Full-time presence of clinical faculty members in public hospitals | | | Identifying and specifying the tariffs and timely payments and establishing special clinics as offices | By strict adherence to the law, may enhance social justice and accessibility to medical services for all members of the community. |
| Dargahi & Zalvand (2019) | Persian | Analysis of physicians' full-time geographical plan by using the SWOT model | Quantitative study | 234 physicians from 15 hospitals | General practitioners in the public sector in a full-time geographical plan | | | Using the TOWS matrix for policy action and analysis is a logical strategy. | Successful implementation of this plan can lead to improved UHC indicators such as reduced payment costs, increased quality and access to services. |
| Etemadian et al. (2020) | Persian | To explore the process of establishing a non-governmental and nonprofit hospital, as a corporate infantry, and a model for the establishment of autonomous and corporate hospitals | Qualitative study | 17 key informants | physicians' dual practice in the public and private sectors | | Financial and earning incentives as the motivators and dissatisfaction with payments as a factor that demotivate physicians | One of the primary factors behind the establishment of Moheb hospital (a nongovernmental hospital in the public sector) was the physicians' dual practice in the public and private sectors. | The establishment of a non-governmental hospital in the public sector increases the quality of and accessibility to services |
| Akbari Sari et al. (2013) | Persian | To explore the perception of the chancellors at Iran universities of medical sciences regarding the challenges, the possible negative consequences of dual practice and collecting their suggestions for improving the implementation of physician's full-time practice program in Iran | Quantitative study (Cross-sectional) | 30 chancellors at Iran universities of medical sciences | The law prohibits the physicians' of the public sector from private practice | | | The need for increasing the share of the healthcare budget from the gross domestic product (GDP), inefficient tariff and payment system, and difficulty in obtaining collaboration with other stakeholders are the main challenges of full-time practice program in Iran. It is suggested that this program should be implemented gradually in a specific set of Iranian hospitals that have already gained more autonomy. | Inappropriate implementation of this program may lead to an unexpected transfer of the experienced and highly skilled physicians of the public and private hospitals. |

(Continued)

**Table 1.** (Continued)

| Author, year | Language | Aim | Study Method | Population | forms | Extent | Predictors, drivers and motivations | policy options | consequences for UHC goals |
|---|---|---|---|---|---|---|---|---|---|
| Akhavan Behbahani & Rahbari Bonab (2015) | Persian | To assess dual practice physicians law | Qualitative study | 15 key informants | Physicians who are employed on a contract or formal basis in public and private non-governmental teaching hospitals are allowed to practice medicine in diagnostic, educational, medical, and private and charitable hospitals. | | | • Developing accurate and comprehensive bylaws<br>• step-by-step and gradual implementation of the law<br>• Paying attention to the financial resources needed for implementation of the law, as well as setting the tariff and an effective and efficient payment mechanism for physicians | • Improving the accessibility to health services<br>• improving the quality of the provided services<br>• reducing costs and increasing the efficiency of the health system<br>• increasing the motivation and skills of physicians |
| Bazyar et al. (2018) | English | To develop policy options (POs) to implement physicians' dual practice prohibition law in Iran | Qualitative study | 14 informed policy makers | Working in both public and private sectors | | | Fifteen policy options were developed and their advantages and disadvantages were discussed. It is suggested to follow a conservative and incremental approach and start with POs that will cause less resistance and political challenges | Implementing these policy options can improve universal health coverage. These policies can increase the quality of services and accessibility to them. |
| Rafiei et al. (2015) | English | To discover neurosurgeons' occupational preferences and propose policy interventions those possibly increase their retention in rural, remote, or underserved areas. | Quantitative study (Discrete Choice Experiment) | 120 neurosurgeons working in public hospitals of 5 Iranian provinces and 15 districts | This means whether physicians are permitted to work in private sector besides public healthcare entities or not | | Providing an opportunity to undertake dual practice may raise the probability of choosing a remote(rural) job by 65% | Permission to undertake dual practice was the dominant policy for increasing the neurosurgeons' retention in rural, remote, or underserved areas | Findings indicated that permission to undertake the dual practice was the most crucial factor rather than other retention policies. |
| Bayat et al. (2018a) | English | To investigate the impact of the dual practice on the service delivery duration by surgeons. | Quantitative study | 4642 surgery specialists | Working simultaneously in the public and private sectors | 63.1% of specialists were engaged in the dual practice i.e. they were working in other service locations as well. | Female specialists and full-time specialists spent less time on health care service delivery. Permanent specialists had higher full-time equivalent (FTE) and as the population increases, FTE increases. | | Dual practice had a direct impact on surgeons' working hours. Specialists with dual practice had long service delivery time. |

*(Continued)*

Table 1. (Continued)

| Author, year | Language | Aim | Study Method | Population | forms | Extent | Predictors, drivers and motivations | policy options | consequences for UHC goals |
|---|---|---|---|---|---|---|---|---|---|
| Palesh et al. (2010) | English | To explore how policy-makers' justify the diffusion and utilization of health technologies in Iran using magnetic resonance imaging (MRI) and interferon beta as tracers. | Qualitative study | 13 informants in different positions and levels of authority in the Ministry of Health (MOH), University of Medical Sciences, Health Insurance Organizations, and Parliament | Physicians working simultaneously in public and private sectors | | lower salaries and income in the public sector compared to the private sector has been identified as a major incentive for dual practice | | • A shift of patients from the public to the private sector<br><br>• enhanced induced demand<br><br>• the reduced ability of government agencies to maintain control |
| Bayati et al. (2019a) | English | To investigate the rate of the dual practice of general practitioners (GPs) as the most important primary care providers in Iran along with identifying the underlying factors | Quantitative study | 666 General practitioners (GPs) | Working in both public and private sectors | 18.65% of the GPs were working in both private and public sectors. 26.17% of the GPs were working in two or more healthcare entities | Gender (males more than females) and earning expectations had a significant effect on the multiple job-holding | | - |
| Bayat et al. (2019) | English | To determine the level of dual practice engagement and its related factors among Iran's surgery specialists. | Quantitative study | 14931 surgeons from 858 hospitals | Attendance of physicians in more than one service delivery location | Overall, 6405 (57%) public sector surgery specialists were engaged in dual practice from which 5060 were in service through MOHME and 70% of non-full time specialists had multiple job holdings. Of 8312 Non-Faculty member specialists, 58% had multiple job holdings. Urology and neurology surgeons with 69% dual practice engagement had the highest proportion | Dual practice was more frequent in specialists with higher age and work experience, populated provinces, more deprived areas, and higher share of private hospitals in service delivery | | - |

(Continued)

**Table 1.** (Continued)

| Author, year | Language | Aim | Study Method | Population | forms | Extent | Predictors, drivers and motivations | policy options | consequences for UHC goals |
|---|---|---|---|---|---|---|---|---|---|
| Bayat et al. (2018b) | English | To identify the causes of medical specialists' tendency towards dual practice in the Islamic Republic of Iran | Qualitative study | 14 key informants | holding multiple jobs at the same time | The rate of DP of public medical specialists in the private sector was 42.3%(18.9% to 69%) | Financial incentives, cultural attitudes about professional identity of physicians, work experience and academic level of specialists, controlling approaches in the public sector, available infrastructure for responding to the population needs in the public sector, and regional characteristics of health service locations. | | - |
| Bayati & Rashidian et al. (2019b) | English | To describe the characteristics of economic behaviors of Iranian GPs | Quantitative study | 666 General practitioners (GPs) | Working in both public and private sectors | | GPs who work in both public and private sectors (dual practice) have much more income than others. | | |
| Bayat et al. (2018) | English | To determine the level and rate of dual practice engagement among Iran's specialists | Quantitative study | 24 414 specialists who worked in Iranian public and private hospitals in 2016 | Holding a job in more than one health care entity simultaneously | 47.7% of specialists were engaged in dual practice | Female specialists had 0.78 times less chance for dual practice; faculties compared to non-faculties had 0.65 times higher chance for dual practice and full-time geographic specialists compared to non-full-time specialists had 0.15 times more chance for dual practice. The dual practice was more frequent in specialists in the higher age and more job experience and in more populous provinces, deprived areas, and a higher number of specialists per facility | | |

services [10, 15]. The highest rates of dual practice were reported in Qazvin (72%), Kohgiluyeh and Boyer-Ahmad (70%), Gilan (69%), East Azerbaijan (59%) and Alborz (57%) provinces [15].

**Motivators and effective factors.**   In nine studies, the motivators and factors affecting dual practice were studied among physicians and specialists [5, 9–12, 14–16, 21]. In five studies, earning incentives and financial incentives were considered the predominant motivators of dual practice; in particular, a lower salary in the public sector compared to the private sector was the main reason for physicians' and specialists' dual practice [5, 9–11, 14]. It was indicated that full-time specialists and female specialists spent less time providing services than full-time general practitioners. Surgeons spent more time in private than public hospitals due to financial incentives and earning more money [12].

Another underlying factor of dual practice was gender. Dual practice rates were higher in males than females; these findings were reported in four studies [10–12, 15].

Results of three studies confirmed that by increasing the individuals' age and work experience, the rate of dual practice also rose [10, 11]. Married people had more dual practice [10]. The probability of dual practice was higher among academic specialists compared to non-academic specialists and higher in geographic specialists compared to non-full-time specialists [15]. In densely-populated and more deprived provinces and in the context in which private sectors participate more in service provision and delivery, the rate of dual practice was significantly higher than that in other studied provinces [10, 15, 16]. Cultural attitudes about physicians' professional identities, work experience and academic level of specialists, public sector control approaches, existing infrastructures to meet the population's needs in the public sector, and regional characteristics of health care entities were among the critical factors affecting dual practice [5, 12].

**Policy options.**   Six studies reported political strategies for addressing dual practice. In Akbari Sari et al.'s study, the principal policy options were identified to be increasing the share of the health sector in GDP and the gradual reform of tariffs and the payment system [20]. In the study by Etemadian et al., the establishment of non-profit and non-governmental hospitals along with public hospitals was mentioned as a viable option to solve the problem of dual practice [9].

In another study by Dargahi et al., SWOT analysis was applied to analyze situations, recognize opportunities, threats, strengths, and weaknesses for policy actions, and design a logical strategy [18]. In Baziar et al.'s study, 15 policy options were developed. Some of these policy options were to implement the policy through the relevant regulations for all physicians at national level. Therefore, in order to enforce the implementation of these laws initially as a pilot program in some provinces of the country, some of these policy options and rules have been only defined for some medical disciplines. For instance, the implementation of these laws has been imposed on some large cities and some specialties, and even in some cases, the laws have been enforced based on faculty members' work experience and recruitment type. Each policy option has been analyzed in terms of advantages and disadvantages [13]. According to one study, the following facilitate the implementation of the law in order to ban the employment of public physicians in the private sector: the development of accurate and comprehensive law enforcement, step-by-step implementation of the law, design, and implementation of a field study to identify various dimensions affecting the prevalence of this phenomenon, showing special attention to the provision of stable and reliable financial resources required for implementing the law and also organizing the tariff and the payment mechanism to physicians for their motivation, improving the status of service provision and delivery in public hospitals in terms of volume and the quality of service delivery, building a promoting culture toward the public hospital services utilization among the people, eliminating mistrust between

the medical community and officials, and last but not least, restrict law enforcement by the executor is among the policy measures of the executors [19]. In another study, the leading policy for enhancing the retention of neurosurgeons in rural areas and remote or deprived provinces was the license to allow them to perform dual practice [16].

**Consequences of dual practice for UHC.**   Nine studies explicitly and implicitly addressed the implications of dual practice on UHC [9, 12–14, 16–20]. Five studies revealed that adherence to the full-time faculty programs [17], full-time geographical programs [18], and prohibition of public physicians from practicing in the private sector [13, 16, 19] can enhance access, the quality of services, and physicians' retention in rural, remote, or deprived areas and reduce out-of-pocket payments. All the above mentioned play a meaningful role in achieving universal health coverage. The study conducted by Akbari Sari et al. reported that the improper implementation of these programs may lead to an increase in the induced demand due to low tariffs, an increase in underpayments, and out-of-pocket payment costs, which all encourage skilled and experienced physicians to move from the public sector to the private one and ultimately reduce UHC [20]. The results of four studies confirmed the effect of dual practice on increasing out-of-pocket payment costs [9, 12–14], and two studies reported a decrease in patients' access to services as a result of dual practice [12, 14].

## Discussion

In the current scoping review, 14 studies on the dual practice of physicians in Iran were included. We tried to determine the research gaps in this area by examining these studies. The present study can help to provide an overview of the phenomenon under study.

The present study indicated that physicians are involved in dual practice in Iran. Most studies have focused on the extent and factors affecting dual practice, and few studies have been conducted on the causes of this phenomenon and policy options to address this issue. Further research is needed for developing a more comprehensive and deeper understanding of the phenomenon of dual practice in Iranian physicians. Therefore, studies in the following fields should be included in the country's research agenda:

First, more research is required at national level on the nature, types, and prevalence of this phenomenon in Iran. This research should be done exclusively for different provinces and medical specialties. In Iran, most studies have focused on the private-public form of dual practice. The dual practice may happen in the form of simultaneous public-public, private-private, or public-private physicians' practice [22]. Therefore, future studies should investigate other forms of dual practice in Iran.

Secondly, future research should focus on clarifying the root causes of this phenomenon and getting a deeper understanding of why physicians engage in dual practice. Such research allows us to describe and quantify the correlation between these factors and the phenomenon under study. It also enables us to understand their relative similarities and differences in the various occupations of the health sector and to explain their effects on the motivation and behavior of health workers. On the other hand, investigating the causes and motivators of physicians' simultaneous practice can help policymakers in formulating the content of policies in this regard by providing a correct map of the root causes of this phenomenon. Needless to say, policies that are designed with a full understanding of the main factors of a phenomenon will be more successful in the implementation phase. Such an understanding can be achieved by using qualitative study designs, especially with grounded theoretical and phenomenological approaches. In the previous studies, numerous factors were mentioned as underlying factors of dual practice, such as higher income, age, gender, marital status, and practice in deprived areas, full-time or part-time activity. In studies conducted in other countries, the rate of dual

practice also varied according to such factors as living in the urban or rural regions, the professional group (in physicians more than other professional groups), the type of specialty, and the type of contracts (Full-time or part-time) [23–26]. Evidence shows that dual practice depends more on managerial factors than personal (age and gender), social (marital status) and occupational characteristics of physicians [27]. The present study also indicated the lack of study on the preferences of physicians in the choice of dual practice and related factors; therefore, future studies should focus on this field. For instance, in a study conducted in Vietnam, more than 60 percent of dual-practice physicians stated that they wished to quit their private practice in exchange for an increase in the basic salary, housing benefits, or promotion opportunities [23]. The results of such studies can be used to design different policy options.

Thirdly, there is a research gap in the effects of dual practice on the indicators of accessibility to health services, especially for vulnerable populations, the quality of care provided, and equity. Most studies carried out on dual practice had not taken advantage of a specific approach for the assessment of its consequences. Obtaining a comprehensive understanding of the impacts of this phenomenon on the intermediate goals of the health care systems, such as access, quality, and equity, is essential for policymaking in this area.

Fourthly, there is a need for complex policy research on health policy options. Future studies should also analyze the previous policies. A comprehensive evaluation should be conducted on the content of laws and regulations and their impacts. On the other hand, the policy options that are formulated and implemented in this subject area are scattered in Iran, and these options should be seen as a "package of interventions" in which economic and non-economic factors are seen because there is no single formula for dealing with the reality of dual practice [27].

In the fifth place, there seems to be a methodological gap (a gap in the type of study and its method) in the subject area. Most studies have only used quantitative or qualitative study methods. It is crucial to use mixed-methods study designs or time series or various research methods to create new insights or to avoid distorting the findings. On the other hand, the applied information in the research, especially such sensitive information as income, was based on the self-report of research samples in most of the included studies. Therefore, more attention should be paid to the accuracy of the collected data in future studies.

Further, the topic investigated here calls for more research for evidence-based decision-making and policymaking appropriate to the context of the phenomenon of dual practice in Iran. This research should seek to understand the complex phenomenon of dual practice in the country's health system and the tools to address it. The research questions discussed above are summarized in Table 2.

The current study is the first attempt to provide an overview of the pieces of evidence in the field of dual practice in Iran. One of the strengths of the study is that the research was conducted and reported based on PRISMA-ScR and codified search strategies for the main databases. This study also faced some limitations. Firstly, the dual practice was specifically studied in physicians as the research sample. There is a need to address the issue of multiple job holding and dual practice in other health service providers such as nurses and primary care professionals. Secondly, given the nature of the scoping review studies [6], we did not assess the quality of the presented evidence.

To apply the results of this study, we suggest that the proposed research questions be provided to graduate students in the fields related to health management, health policy, and health economics. Policymakers can also be aware of the results of this study to seek support and apply the results of this study in formulating reforms in the health system in order to achieve universal health coverage.

**Table 2. Suggested questions in the field of physicians' dual practice in Iran for conducting future studies.**

| Main issues | Possible research questions |
|---|---|
| Extent and nature of dual practice | • What is the rate and forms (simultaneous public-public, private in public or the public in private) of the dual practice of physicians in the country? |
| | • What are the differences in physicians' practice between the public and private sector in the Iranian health system? |
| Predictors, Drivers and Motivators | • What individual, professional, organizational, and contextual factors are involved in the extent of dual practice? |
| | • What are the physicians' preferences in working in the public and private sectors? Which attributes affect it more? |
| Consequences of the dual practice | • How does this phenomenon affect the performance, job satisfaction, and motivation of physicians? |
| | • What are the effects of the dual practice and the allocation of physicians' time to both public and private sectors on physicians' incomes, the national health systems, waiting times for services, patients' costs, and their health consequences? |
| | • What are the effects of the phenomenon of dual practice on physicians' burnout? |
| | • What are the effects of dual practice on the induced demand for diagnostic and inpatient services? |
| Differences by type of specialists | • What are the differences in dual practice among different medical specialties? |
| | • What are the reasons for such differences? |
| Differences within the country | • What are the differences in dual practice among various provinces and cities of the country? |
| | • What are the reasons for such differences? (For example, practice in the urban or rural area, different socio-economic, cultural, political or foreign contextual factors) |
| Comparison with other countries | • What is the situation of Iran in the extent and forms of dual practice in the health system compared to other countries? |
| | • What factors play a key role in this phenomenon? (Such as the structure of the health system, its centralization or fragmentation, the role of the private sector, the existence of laws and policies in this area, etc.) |
| | • How do the legislation and organizational infrastructure of dual practice vary in Iran compared to other countries? |
| Designing policy interventions | • What are the strategies for improving the management and compensation of physicians in the public sector? |
| | • What are the disadvantages and advantages of different policy options implemented in Iran, such as private practice in public hospitals (Moheb Hospital experience)? |
| | • What are the challenges and obstacles of dual practice policy in Iran in terms of policy formulation, implementation, and evaluation? |
| | • What are the consequences of policies, interventions, and laws related to dual practice in Iran? (Policy evaluation studies) |

## Conclusions

The review demonstrated the necessity of paying considerable attention to the issue of the dual practice of physicians in Iran. Investigating the country-level contextual factors is crucial for achieving a more comprehensive understanding of the phenomenon, the underlying factors, and their consequences. The policy options that are formulated and implemented in this area are scattered. The factors affecting this phenomenon are intertwined. Dual practice policy should be considered an integral part of the health human resources strategy in the country. Eventually, without such an inclusive perception and taking into account the complexity of the phenomenon of dual practice, policymakers are not able to design and implement a comprehensive policy in this area. Ultimately, there will be a hard way of improving the performance of the health system and achieving universal health coverage.

## Supporting information

**S1 Table. Search strategies and results for selected databases.**
(DOCX)

**S1 Checklist. Preferred Reporting Items for Systematic reviews and Meta-Analyses extension for Scoping Reviews (PRISMA-ScR) checklist.**
(DOCX)

## Acknowledgments

The authors of this paper would like to thank the officials of the National Institute of Health Research of I.R.Iran, Tehran University of Medical Sciences.

## Author Contributions

**Conceptualization:** Javad Moghri, Jalal Arabloo.

**Data curation:** Jalal Arabloo, Mohammad Barzegar Rahatlou, Maryam Saadati, Negar Yousefzadeh.

**Formal analysis:** Jalal Arabloo.

**Funding acquisition:** Jalal Arabloo.

**Investigation:** Javad Moghri, Jalal Arabloo, Mohammad Barzegar Rahatlou.

**Methodology:** Javad Moghri, Jalal Arabloo, Mohammad Barzegar Rahatlou, Maryam Saadati.

**Project administration:** Javad Moghri, Jalal Arabloo.

**Supervision:** Javad Moghri, Jalal Arabloo.

**Writing – original draft:** Jalal Arabloo.

**Writing – review & editing:** Javad Moghri, Mohammad Barzegar Rahatlou, Maryam Saadati, Negar Yousefzadeh.

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
