## [Decision Letter · Decision Letter 0]

27 Oct 2022

PONE-D-22-23688Current knowledge of Physicians' dual practice in Iran: A scoping review and defining the research agenda for achieving Universal Health CoveragePLOS ONE

Dear Dr. Arabloo,

Thank you for submitting your manuscript to PLOS ONE. After careful consideration, we feel that it has merit but does not fully meet PLOS ONE’s publication criteria as it currently stands. Therefore, we invite you to submit a revised version of the manuscript that addresses the points raised during the review process.

We look forward to receiving your revised manuscript.

Kind regards,

Andrea Cioffi

Academic Editor

PLOS ONE

Journal Requirements:

Additional Editor Comments:

The manuscript is well written, however - as indicated by the reviewer - some minor revisions are necessary.

Reviewers' comments:

Reviewer's Responses to Questions

**Comments to the Author**

1. Is the manuscript technically sound, and do the data support the conclusions?

Reviewer #1: Yes

2. Has the statistical analysis been performed appropriately and rigorously? 

Reviewer #1: Yes

3. Have the authors made all data underlying the findings in their manuscript fully available?

Reviewer #1: Yes

4. Is the manuscript presented in an intelligible fashion and written in standard English?

Reviewer #1: Yes

5. Review Comments to the Author

Reviewer #1: Reviewer’s Comment

I am pleased reviewing this article. I found it interesting and informative for policymakers and researchers. The authors have used a sound methodology and rigorously searched for available evidence. In general, it is a well-designed study, used scientific research methodology and written in standard English. I have only few comments for the authors

Abstract

Method(L 28-33): As inclusion criterion, did you use a time frame for studies to be included in your review? If so, I suggest including the timeframe, otherwise mention that there was no time frame used as a searching strategy. Usually a certain period of time is included to narrow your searching strategies so that most relevant studies are included.

Result(L 36-47): The result and conclusion section of the abstract needs a minor revision to make a smooth flow and coherent. In my opinion, it would be much better if you add summary of your findings before you put your recommendation about "more research is required...". Therefore, L39-43 seems a recommendation and need to be moved into conclusion. Similarly, under result section you can add your main findings which are basically limited evidence on the subject matter and methodological gaps even for those identified few articles.

Materials and methods

1) I see you have mention certain inclusion criteria. It is also important to list out you exclusion criteria. I recommend explaining the inclusion and exclusion criteria explicitly.

2) Why didn’t you add a third reviewer when there was a disagreement besides of the in-depth discussions? I believe using a third reviewer can be tie breaker during those disarrangements.

6. PLOS authors have the option to publish the peer review history of their article (what does this mean?). If published, this will include your full peer review and any attached files.

Reviewer #1: No

---

## [Author Response · Author response to Decision Letter 0]

28 Oct 2022

Additional Editor Comments:

The manuscript is well written, however - as indicated by the reviewer - some minor revisions are necessary.

Response: Thank you for your consideration.

Reviewer #1: Reviewer’s Comment

Comment: I am pleased reviewing this article. I found it interesting and informative for policymakers and researchers. The authors have used a sound methodology and rigorously searched for available evidence. In general, it is a well-designed study, used scientific research methodology and written in standard English. I have only few comments for the authors

Response: Thank you for your consideration and valuable comments. 

Abstract

Method(L 28-33): As inclusion criterion, did you use a time frame for studies to be included in your review? If so, I suggest including the timeframe, otherwise mention that there was no time frame used as a searching strategy. Usually a certain period of time is included to narrow your searching strategies so that most relevant studies are included.

Response: Thank you. We added "until August 3, 2020" to the method sections of abstract and main text. 

Result(L 36-47): The result and conclusion section of the abstract needs a minor revision to make a smooth flow and coherent. In my opinion, it would be much better if you add summary of your findings before you put your recommendation about "more research is required...". Therefore, L39-43 seems a recommendation and need to be moved into conclusion. Similarly, under result section you can add your main findings which are basically limited evidence on the subject matter and methodological gaps even for those identified few articles.

Response: Thank you. We moved L39-43 into conclusion section. We also added following sentences to the result section: There were limited evidence on the nature, types, and prevalence of this phenomenon for different provinces and medical specialties and on health policy options in Iran. There seems to be a methodological gap (a gap in the type of study and its method) in the subject area. Most studies have only used quantitative or qualitative study methods and based on the self-report of research samples in most of the included studies. 

Materials and methods

1) I see you have mention certain inclusion criteria. It is also important to list out you exclusion criteria. I recommend explaining the inclusion and exclusion criteria explicitly.

Response: Thank you. We corrected this section as follow: 

The studies were selected based on the following eligibility criteria:

Inclusion criteria: 

1) All primary or secondary studies on dual practice in physicians at any level of the Iranian health system 

2) Only the studies with a full text in English or Persian were included.

Exclusion criteria: 

1) Studies on dual practice in other health professionals such as nurses 

 2) Review studies, editorials, letters to the editors, abstracts, and commentaries; 

3) Studies without a full text or with a full text in a language other than English or Persian; 

4) Duplicate studies whose findings have been published in several articles, and in this case, the most high-quality ones were included in the study.

2) Why didn’t you add a third reviewer when there was a disagreement besides of the in-depth discussions? I believe using a third reviewer can be tie breaker during those disarrangements.

Response: Thank you. We corrected this section as follow: Disagreements between reviewers were resolved by the decision of a third reviewer.

---

## [Editor Report · Decision Letter 1]

6 Nov 2022

Current knowledge of Physicians' dual practice in Iran: A scoping review and defining the research agenda for achieving Universal Health Coverage

PONE-D-22-23688R1

Dear Dr. Arabloo,

We’re pleased to inform you that your manuscript has been judged scientifically suitable for publication and will be formally accepted for publication once it meets all outstanding technical requirements.

Kind regards,

Andrea Cioffi

Academic Editor

PLOS ONE
---

## [Editor Report · Acceptance letter]

9 Nov 2022

PONE-D-22-23688R1 

Current knowledge of Physicians' dual practice in Iran: A scoping review and defining the research agenda for achieving Universal Health Coverage 

Dear Dr. Arabloo:

I'm pleased to inform you that your manuscript has been deemed suitable for publication in PLOS ONE. Congratulations! Your manuscript is now with our production department. 

Kind regards, 

on behalf of

Dr. Andrea Cioffi 

Academic Editor

PLOS ONE